# Control of *Aculops lycopersici* with the Predatory Mite *Transeius montdorensis*

**DOI:** 10.3390/insects13121116

**Published:** 2022-12-03

**Authors:** Cristina Castañé, Oscar Alomar, Alfred Rocha, Enric Vila, Jordi Riudavets

**Affiliations:** 1IRTA, Sustainable Plant Protection, Ctra. Cabrils Km 2, 08348 Cabrils, Barcelona, Spain; 2AGROBÍO S.L., Ctra. Nacional 340, Km 419, El Viso, 04745 La Mojonera, Almería, Spain

**Keywords:** tomato russet mite, eriophyid mite, Phytoseiidae mite, tomato crop, biological control

## Abstract

**Simple Summary:**

The tomato russet mite (TRM) is a significant problem in greenhouse tomato crops, and it has recently become more prevalent in Europe. The few acaricides available for its control are not specific and their effectiveness is limited. Several predatory mites have been tested for regulating TRM populations. Although these predators consumed most TRM stages in lab conditions, they were not able to freely move across the tomato plant due to the presence of glandular trichomes. TRM seeks refuge from predation precisely in the dense layer of glandular trichomes, in the upper part of the tomato plant. Consequently, most of predatory mites failed under crop conditions due to a lack of adaptation to the tomato plant. In this study, the predatory mite *Transeius montdorensis* has shown that it is adapted to the tomato plant since it was able to move through the trichome layer and prey upon TRM. When *T. montdorensis* was preventively released together with its breeding prey, it was able to greatly limit the growth of the TRM population in a tomato plot, and thus an increase of yield was observed. Therefore, *T. montdorensis* is a promising candidate for the control of TRM.

**Abstract:**

In this study, the predatory mite *Transeius montdorensis* (Acari, Phytoseiidae) was tested for the control of the tomato russet mite (TRM) *Aculops lycopersici* (Acari, Eriophyidae) in experiments with small plants, under semi-field and crop conditions. The releasing strategy consisted of repeatedly introducing the predator together with additional breeding prey. The predator was able to move and disperse to the upper part of the tomato plant where the TRM seeks refuge. At the crop level, significant reductions in TRM populations were observed that resulted in a significantly higher yield compared to the conventional control plot, where pesticides were used to control the pest. Caution should be taken when extreme temperatures or humidity occur as they could be deleterious to the predator population. Hence, crop practices should include the management of environmental parameters in the greenhouse to ensure the success of this TRM-control strategy. In conclusion, this biological approach seems to be an effective measure to control the pest and should be further implemented at crop level.

## 1. Introduction

The tomato russet mite (TRM) *Aculops lycopersici* (Tyron) (Acari, Eriophyidae) is a key pest in field tomatoes that causes significant losses [1]. In greenhouses, TRM reaches even higher densities than in field crops due to its adaptation to high temperatures [2]. In Europe, this pest is an increasing problem in both heated (in the north) and unheated (in the south) greenhouses [3].

This mite damages tomatoes by feeding on the epidermal cells of the leaves and the stems, causing a reduction in photosynthesis in the damaged tissue [4,5]. In greenhouse conditions, due to its exponential growth, the mite can dry up to 79% of the leaves and kill the plant [6]. Depending on the infestation level, yields may be reduced by 25% to 100% [7]. With large infestations, russeting may occur in young fruit, reducing the crop’s commercial value [8].

It is difficult to sample this pest because of its small size (140–300 μm long) [9]. By the time symptoms are observed in the plants (a brownish color in the stems and petioles due to the collapse of glandular trichomes), the pest population is already quite large [6]. At this point, it is difficult to control the pest’s spread. Few pesticides are authorized for controlling this pest; sulfur, abamectin, spirotetramat, and spiromesifen can provide acceptable levels of control [3]. However, their continued use creates resistance problems, reducing the effectiveness of the products in the medium term. Therefore, alternative control methods are needed.

Several species of predatory mites can feed on TRM, as several authors have demonstrated in laboratory experiments. However, their efficacy in the field is quite limited [3]. One reason for this is that many predatory mite species cannot move easily through the bush of the glandular trichomes of the tomato plants, while the TRM moves freely in those areas due to its smaller size [10]. Therefore, while predation of TRM by predatory mites is effective in the parts of the plant where the trichomes have been collapsed by the action of the mite, it is not effective in areas with intact glandular trichomes, as TRM can escape predators by seeking refuge in these areas [11].

*Transeius montdorensis* (Schicha) (Acari, Phytoseiidae), a polyphagous predatory mite originally found in New Caledonia [12], effectively consumes several pests (i.e., thrips, eriophyid, and tetranychid mites) [12,13], and it is commercially available. It is a promising candidate for the control of TRM since a preliminary trial on potted tomato plants showed successful control of the pest. However, this was true only when high populations of *T. montdorensis* were maintained on the plants by adding astigmatid mites as alternative food for the predator (E. Vila, data not published). Therefore, the objective of the present study was to test the following hypothesis: if a large predator population is maintained in the crop by supplying the breeding prey, *T. montdorensis* would be able to control or reduce the damage caused by TRM. The aim was to develop a preventive strategy for controlling *A. lycopersici* by introducing *T. montdorensis* before the arrival of the pest and periodically adding astigmatid mites to maintain a high predator population on the plant.

## 2. Materials and Methods

### 2.1. Mite Colonies

A laboratory colony of *A. lycopersici* was initiated with individuals collected from a tomato crop in Calella (Barcelona, Spain) and was maintained on tomato plants at a controlled temperature (25 ± 1 °C; 70 ± 10% relative humidity (RH)). *Transeius montdorensis* and the astigmatid breeding prey (*Suidasia medanensis* Oudemans, Acari: Astigmata) were provided by Agrobío SL.

### 2.2. Experiment in Potted Plants

A test with tomato plants (8–10-leaf stage, var. Caniles, Zeraim Ibérica S.A., Almería, Spain) was performed under controlled conditions (25 ± 1 °C; 70 ± 10% RH). The densities of TRM in the presence and absence of predators were compared. Eight plants per treatment were used in ventilated cages. For the predator treatment, *T. montdorensis* (7 mL/plant, approx. 170 ind.) was released one time together with astigmatid mites (7 mL/plant of *S. medanensis*, approx. 140,000 ind.). Ten days later, sections (1–3 cm long) of tomato stems infested with TRM (~100 individuals/section/plant) were introduced at the bottom of each plant. After this, astigmatid mites (7 mL/plant) were released biweekly five times over a three-month-period, until the plants had 16 to 18 leaves. In the control treatment, the same TRM releases were conducted, but no predator was released.

The total mite densities (pest and predator) were assessed via three fortnightly destructive samplings (one plant per treatment in the first, and three plants in the second and third). The plants were cut into pieces that were then soaked in a wetting solution (distilled water + 0.1% Tween) for 30 min to allow the mites to detach from the plant. The solution was then filtered (20 µm sieve), and the filtrate was redissolved in 60 mL of wetting solution. Ten aliquots of 1 mL of this wetting solution were then counted under a stereomicroscope (Stemi 508, Carl Zeiss microscopy, 37081 Gottingen, Germany) (25×). In addition, on eight plants per treatment a magnifier lens (10×) was used to register every fortnight the maximum height at which collapsed trichomes (damage limit) and the predator were found on the stem. The total height of the plant was also annotated.

### 2.3. Experiment in Cages

A test in an experimental greenhouse (430 m^2^) with 12 large exclusion cages (6.40 × 4.50 m each, netted with a 10 × 20 thread/cm^2^ plastic mesh) was conducted from September 2018 to February 2019. Each cage had five standard coconut fiber growing bags with two tomato plants (four weeks old, var. Caniles) per bag for a total of ten plants per cage. The plants were tied to string, pruned as needed, and maintained on a fertigation system. Temperature and relative hygrometry were recorded using a digital data logger (Testo 175H2, Instrumentos Testo, 08348 Cabrils, Barcelona, Spain) placed 2.15 m above ground level in the middle of the greenhouse.

Two treatments, one with predators and one control (without predators), were tested using a randomized complete block design with six replicates per treatment. Three weeks after plantation, the predator was introduced at the bottom of the plants for the predator treatment (~350 individuals per plant), together with the additional breeding prey (~70,000 individuals per plant). The prey was sprinkled on the top of plants to force the predators to distribute evenly across the plants. One week later, all plants in both treatments were artificially infested with the pest. TRM was introduced by placing an infested piece of stem containing approximately 100 individuals from the laboratory colony at the bottom of each plant. To feed the predator population, five more releases of breeding prey (~70,000 individuals per plant in each one) were conducted every fortnight.

The plant phenology and mite population density were assessed fortnightly (six samplings) by inspecting three randomly selected plants in each cage. The number of leaves and the youngest leaf with collapsed trichomes (damage limit in height) on each plant were assessed with the aid of a magnifier lens (10×). The numbers of TRM and *T. montdorensis* in the upper and the lower leaves around the damage limit (top and bottom sides inspected) were also evaluated. This was done by sticking and removing 1 cm^2^ tape sections (as in [14]) to the petiole junctions of these leaves with the stem (two tape sections per plant). In total, two tapes per plant were collected at each sampling date. Afterwards, these tapes were attached to a transparent acetate sheet and examined under a stereomicroscope (25×) for the presence and abundance of TRM and/or *T. montdorensis*. To assess pest damage at the top of the plants, where the damage limit was located, a severity index was used: (0) no damage, (1) very small spots or trails of collapsed trichomes, (2) large spots of collapsed trichomes, and (3) collapsed trichomes around the entire perimeter of the stem.

Mite densities were also estimated via five destructive samplings taking one plant on each cage. For each sampling, two leaves above and two leaves below the damage limit of TRM were cut and soaked for 30 min in water containing a wetting agent (0.1% Tween) to shed the mites. The solution was then filtered through a 20 µm light sieve, and the residue of the filtrate was resuspended in 60 mL of 0.1% Tween solution. Five aliquots of 1 mL per plant were then counted under a stereomicroscope (25×).

### 2.4. Experiment in a Greenhouse

Two tests, one in the autumn cycle (July–December 2020) and another in the spring cycle (February–August 2021), were performed in an experimental unheated plastic greenhouse (416 m^2^). The greenhouse was divided in two plots using Agryl mesh, and 240 tomato plants (var. Anairis grafted on the rootstock Caramba, De Ruiter, The Netherlands) were planted in each plot at a density of 2.4 plants/m^2^. To facilitate correct pollination, one bumblebee hive was introduced into each plot. The plants were tied to strings, pruned as needed, and maintained on a fertigation system. An integrated pest management (IPM) program based on conservation of predatory mirid bugs was used to control pests in both plots. This IPM program is regularly used by farmers in the area around Barcelona (Spain).

Each test included two treatments: one with predator and one control (no predator). In the control plots, treatments to control TRM recommended by pest-control advisers were used. In the predator plots, *T. montdorensis* and its breeding prey were released with an air gun dispenser. In the autumn, five releases of the predator (for a total of 495 predators per plant) together with its breeding prey (for a total of 112,500 preys) were performed; seven releases of the predator (for a total of 885 predators per plant) and its breeding prey (for a total of 212,500 preys) were done in the spring trial. The mite density was assessed weekly via inspections of fifteen randomly selected plants from each plot. To record the number of TRM and *T. montdorensis* on each inspected plant, two to five (based on the height of the plant) sticky-tape sections (1 cm^2^) were attached to the plant stems. These tapes were examined under a stereomicroscope (25×) for the presence and/or the number of trapped individuals of TRM and/or *T. montdorensis*. The total number of fruits produced, the number of commercial-quality fruits, and the weight of the fruits were evaluated for ten randomly selected plants from each plot.

### 2.5. Data Analyses

In the cages experiment, mite-days were calculated (the mean of two consecutive counts multiplied by the number of days between those two counts) for TRM and *T. montdorensis*. Statistical comparisons were then performed.

A Student’s *t*-test was used to compare treatments per sampling date. Data from the sticky tapes and from the destructive samplings of mites in the two treatments (with and without predators) were compared, as were the number of plant leaves, the maximum height of plant damage, and the severity of damage at the end of the experiment. Data from the sticky-tape samplings in the greenhouse experiment were also compared by this test.

A paired *t*-test was used to compare the number and weight of total and commercial-quality fruits from each harvesting date, between the two treatments (with and without predators) in the greenhouse experiment.

All statistical analyses were conducted using JMP^®^ (version 14.2.0, SAS Institute, Cary, NC, USA) [15].

## 3. Results

### 3.1. Experiment in Potted Plants

In the control treatment, the total number of TRM was quite high, especially on the two last sampling dates (13,000 to 16,000 individuals estimated per plant) (Figure 1). However, in the predator treatment, russet-mite levels were low (less than 10 individuals and 0 individuals per plant on the last sampling date) while predator levels were high (maximum of 200–450 individuals per plant) (Figure 1).

The maximum height at which collapsed trichomes were found was similar in both treatments, reaching the top of the plant by the end of the test (Figure 2). Predators were found throughout the plants, including the area above the maximum height of collapsed trichomes in areas where the glandular trichomes were still intact. On the last sampling date, collapsed trichomes were found on top of the plants in both treatments, although there were no TRM in the predator treatment, as shown in Figure 1. It seems that predators began to disappear due to the lack of TRM as prey.

### 3.2. Experiment in Cages

During the test, the mean temperature was 18 °C (range: 9–28 °C), and the mean RH was 69% (range: 38–85%).

When the abundance of TRM was evaluated using sticky tape, significantly more mites were found in the control than in the predator treatment in the last sampling dates (mite-days: *t* = −3.75; *df* = 3.150; *p* = 0.030) (Figure 3). A similar trend in the TRM population was observed via destructive sampling, although the numbers were several orders of magnitude higher. However, in the final destructive assessment, the difference between the treatments was not statistically significant (mite-days: *t* = −2.11; *df* = 3.17; *p* = 0.120). The predator population, as measured by sticky-tape samples, remained stable until early November, when it began to decline until it disappeared in mid-December. In December, average temperatures were 12.8 ± 0.28 °C; the shortest photoperiod of the year also occurs then. The destructive samples of the predator population revealed the same trend, but these numbers were two orders of magnitude higher than those of the sticky tapes. This destructive sampling verified the reproduction data about the predator: from the approximately 350 individuals released per plant on the 11th of October, a mean of 3045 mite-days (mean ± SE) per four leaves was recorded one month later, from the 12th to the 26th of November.

At the end of the test, the plants in the two treatments had similar numbers of leaves (*t* = −1.02; *df* = 6; *p* = 0.347), but the maximum height of collapsed trichomes was higher in the control than in the *T. montdorensis* treatment (*t* = −3.06; *df* = 6; *p* = 0.022) (Figure 4). The severity damage index was also significantly higher in the control than in the *T. montdorensis* treatment (Figure 4) (*t* = −5.74; *df* = 6; *p* = 0.001).

### 3.3. Experiment in a Greenhouse

In the autumn cycle, the first TRM foci appeared four weeks after transplanting, early enough to hinder any preventive strategy. Two treatments, one with wettable-sulfur and another with spiromesifen, were used to reduce the mite population before the predator releases were started. Four wettable-sulfur treatments, three spiromesifen treatments, and one abamectin treatment were used to control TRM in the control plot, while two wettable-sulfur treatments, one spiromesifen treatment, and five releases of *T. montdorensis* with their breeding prey were used in the predator plot. In addition, two treatments with *B. thuringiensis* were used to control *Helicoverpa armigera* Hübner in both plots; one indoxacarb treatment was used to control *Liriomyza* sp. in the control plot, and one release of the parasitoid *Diglyphus isaea* (Walker) was done in both plots.

In the spring cycle, the first TRM foci appeared at the beginning of May, ten weeks after transplanting. Predator releases started as soon as the first TRM focus was detected in the periodic samplings. Therefore, no preventive strategy was used since the predator population was not yet established before the first TRM was detected. To control TRM, four wettable-sulfur treatments were used in the control plot, while one wettable-sulfur treatment and seven releases of *T. montdorensis* with its breeding prey were done in the predator plot. In addition, to control powdery mildew, three wettable-sulfur treatments were conducted in the control plot, and three fungicide treatments (difenoconazole, metrafenone, and azoxystrobin) were conducted in the predator plot. More predator releases were done than planned because in mid-June, after four predator releases, a few very dry days (mean humidity below 52–65%) killed the predator population. The releases were then re-initiated.

The spread of the pest across plants (based on the presence of TRM trapped on the sticky tapes) was similar in both plots; the pest spread quicker in the *T. montdorensis* plot during the spring cycle (Figure 5). This confirms that the predator did not prevent the spread of the pest within a single plant (as shown in the experiment in potted plants) or between plants (as shown in the experiment in cages). That is, the predator did not stop the dispersal of the pest but, rather, reduced its density.

The density of individuals trapped on the sticky tapes at the end of the autumn crop was significantly lower in the predator plot than in the control (*t* = −2.22; *df* = 19.8; *p* = 0.038). On the 27th of October, the mean TRM was also higher on the control than in the predator plot, although this difference was not statistically significant (*t* = −1.76; *df* = 19.2; *p* = 0.096). Similar results were obtained during the last month of the spring crop—the density of individuals on the sticky-tape samples in both plots was similar at the first two counts (1st and 7th July) (*t* = 0.75; *df* = 27; *p* = 0.460 and t = −0.66; *df* = 16.7; *p* = 0.519, respectively), but it was significantly lower in the *T. montdorensis* plot at the last two counts (15th and 21st July) (*t* = −2.65; *df* = 14.8; *p* = 0.018 and *t* = −2.85; *df* = 14.5; *p* = 0.012, respectively) (Figure 5). Plants had a greener aspect in the biological than in the control plot at the end of the autumn cropping season (Figure 6).

In the autumn crop, the yield of predator plot was significantly higher than that of the control plot (total number of fruits: *t* = 2.406; *df* = 13; *p* = 0.031; total fruit weight: *t* = 4.303; *df* = 13; *p* = 0.0009; commercial fruit weight: *t* = 5.113; *df* = 13; *p* = 0.0002). However, in the spring crop, only the number of fruits in the two plots differed significantly; differences in fruit weight were not significant (total number of fruits: *t* = 2.172; *df* = 15; *p* = 0.043; total fruit weight: *t* = 1.415; *df* = 15; *p* = 0.178, commercial fruit weight: *t* = 0.912; *df* = 15; *p* = 0.376) (Figure 7).

## 4. Discussion

*Transeius montdorensis* was originally found by Schicha feeding on eriophyid mites in tomato and other plants [12]. Unlike other predatory mites, *T. montdorensis* would be naturally adapted to the tomato plant, which explains why, in the potted-plants test, the predator was able to disperse to the upper part of the plant, where the trichomes were not yet collapsed by the action of the TRM. Despite this, the predator was not able to limit the dispersal of the eriophyid mite upwards on the plant, since damaged trichomes were found at similar heights in both treatments. The collapse of trichomes in tomato plants is a mechanical and physiological reaction to the TRM’s feeding activity. A massive trichome collapse was observed wherever TRM numbers increased beyond a threshold of approximately 50 mobiles per cm^2^. TRM can easily reach this number within days after a moderate initial contamination [11]. Nevertheless, *T. montdorensis* was able to limit the growth of the eriophyid population, and eventually eradicate it in some settings such as that of the first experiment (Figure 1), in spite of two different tomato varieties being used.

*Transeius montdorensis* also effectively limited TRM growth in the cages experiment. The predator was able to reduce the damage to the plants, reducing both the height at which collapsed trichomes were found and the severity of the damage. The predator population was maintained and grew at the expense of the astigmatid prey, and since it is a polyphagous predator, their large populations also consumed great quantities of TRM. However, the environmental parameters for the predator’s survival and reproduction are narrower than those of the pest. For this reason, from mid-November onwards, low temperatures and short photoperiods probably favored the decline of the predator population, while these environmental conditions still supported TRM development.

The numbers of TRM were very high on the final sampling dates. The sticky-tape sampling method and the leaf-washing method revealed similar trends in predator and pest populations in the cages experiment, suggesting that the sticky-tape method provided a good estimate of the populations of both mites.

During the autumn tomato cycle, the releases of *T. montdorensis* limited TRM density enough to significantly affect yield since there were no differences in other pests or diseases between plots that could account for this yield variation. This positive impact occurred even though we did not employ the preventive strategy of installing the predator before TRM appeared on the crop. In the spring cycle, we had to start new releases of the predator in mid-June, after four previous releases, due to harsh environmental conditions that killed the predator. In the spring crop, the yields for the two treatments differed only in the number of fruits. According to [2] a heavy infestation of *A. lycopersici* affects plant height, the number of leaves, and the diameter of the stem in plants—as much as 60% to 80% of leaves may be brown and dry in infested plants. In addition, according to [2], as TRM densities increased, individual mites increased their feeding activity. One possible explanation is that, as mite density increased, the mites moved more often to escape crowding. This increased mite movement resulted in increased probing by each individual, thereby increasing the number of damaged cells [4]. This led to a reduction in photosynthesis, which was reduced by 50% per 450 mite-days per cm^2^ [5]. Mite-days higher than this were observed in the present study (estimated 800–1900 mite-days/cm^2^ in the controls of the cages experiment). Although mite feeding causes no direct damage to the mesophyll, the probable source of this reduction in photosynthesis is the destruction of the guard cells located amidst the epidermal cells [4]. The destruction of guard cells results in the closure of stomata, severely inhibiting leaflet gas exchange and subsequent photosynthesis.

## 5. Conclusions

In a commercial crop, *T. montdorensis* was able to limit the densities of the TRM when the predator population was maintained through the periodical release of its breeding prey. A preventive strategy for establishing and maintaining the predator population should be optimized to improve TRM control; this strategy should also be adapted to the different environmental conditions of crops in different regions. Alternative prey should be released to allow the predator population to thrive, and these releases should be more precisely fine-tuned to make it more economically feasible. Moreover, different commercial tomato varieties may differ in the abundance of the various types of trichomes which can have an influence on the predator performance. In addition, the Astigmata species to be supplied as prey should be considered carefully since *S. medanensis* has been cited as a potential agent causing allergenic reactions in humans.

## Figures and Tables

**Figure 1 insects-13-01116-f001:**
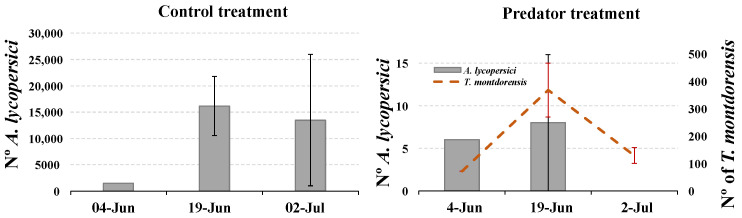
Number of *A. lycopersici* and number of predators per plant in the control and in the predator treatment. Scores were obtained via three destructive plant-washing samplings; the density of the collected mites was measured using a stereomicroscope (25×).

**Figure 2 insects-13-01116-f002:**
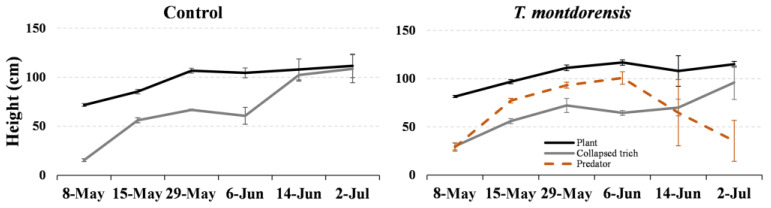
Height of the plant, maximum height at which collapsed trichomes were found, and maximum height at which *T. montdorensis* was found on each sampling date for the control and predator treatment.

**Figure 3 insects-13-01116-f003:**
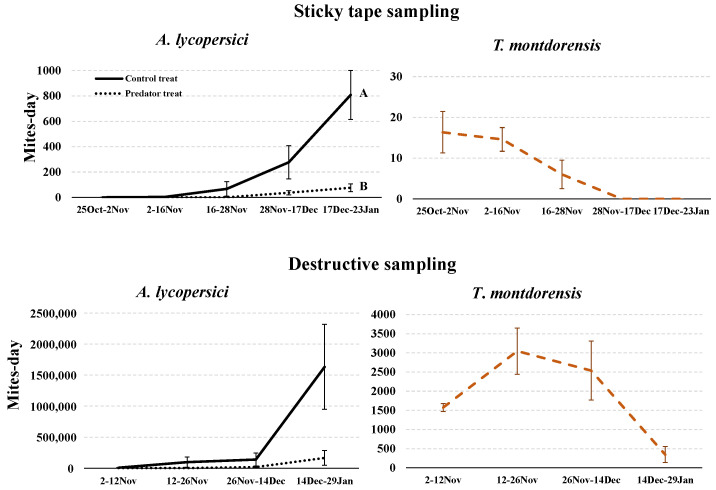
Densities of TRM and *T. montdorensis* (mite-days) per four leaves in the experiment with cages, autumn 2018. Counts were made using sticky tape or destructively by washing four leaves; the density of the collected mites was measured using a stereomicroscope (25×). Significant differences between treatments are indicated by capital letters (*p* < 0.05).

**Figure 4 insects-13-01116-f004:**
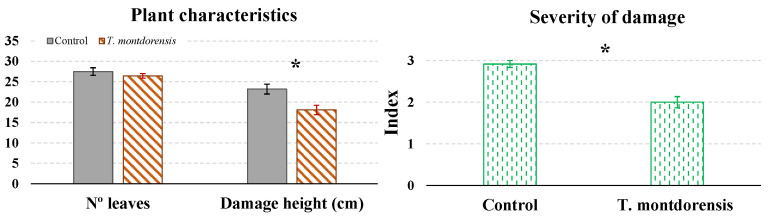
Number of leaves and maximum height at which trichomes collapsed by TRM in the control and predator treatments on the last sampling date. The figure also shows the severity of the damage in the upper part of the stem: (0) no damage, (1) very small spots or trails of collapsed trichomes, (2) large patches of collapsed trichomes, and (3) collapsed trichomes around the entire perimeter of the stem. Black asterisks indicate significant differences between treatments (*p* < 0.05).

**Figure 5 insects-13-01116-f005:**
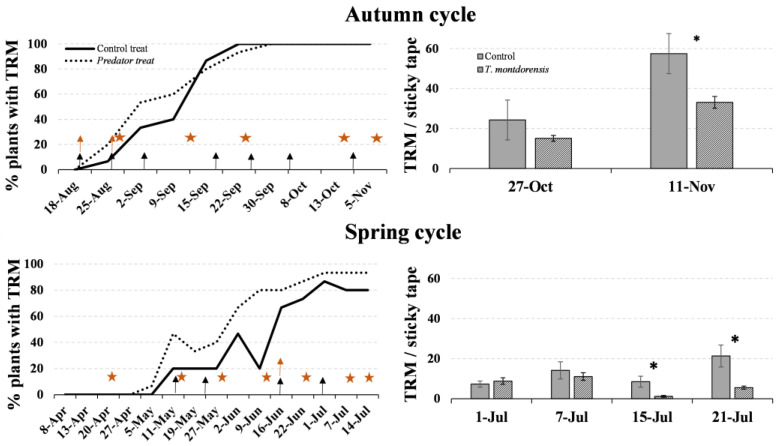
Percentage of plants with TRM and *A. lycopersici* density, measured using sticky tapes in two greenhouse tomato crop cycles, autumn 2020 and spring 2021. Arrows indicate the phytosanitary treatments against *A. lycopersici* that were used in each plot (black for the control and yellow for the predator treatment); yellow asterisks indicate predator releases. Black asterisks indicate significant differences between treatments (*p* < 0.05).

**Figure 6 insects-13-01116-f006:**
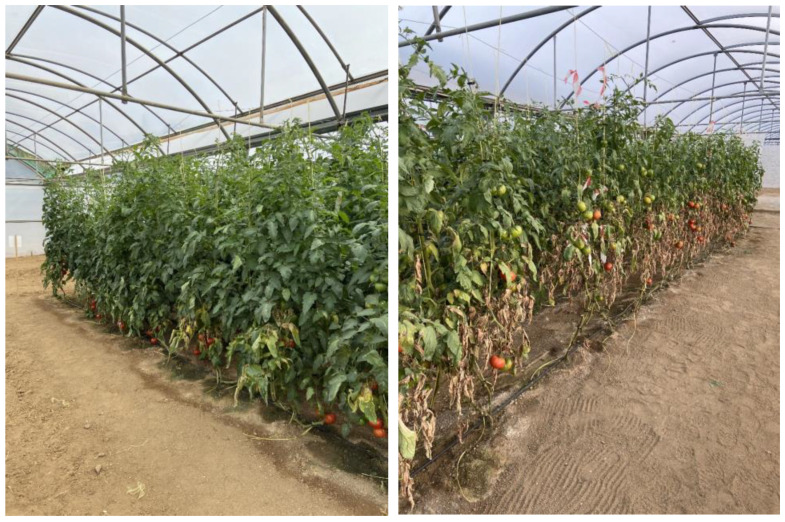
Appearance of plants in the biological (**left**) and control plot (**right**) at the end of the autumn crop.

**Figure 7 insects-13-01116-f007:**
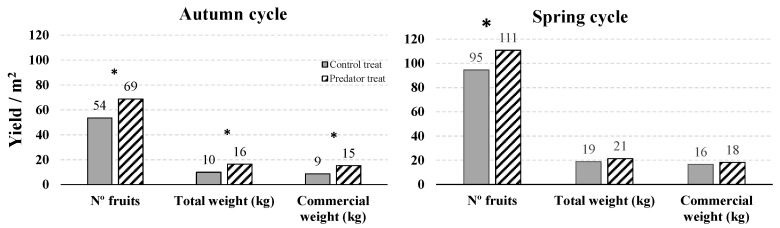
Number of fruits, total fruit weight, and weight of commercial fruits in the two treatment plots for the two crop cycles performed. Black asterisks indicate significant differences.

## Data Availability

The data presented in this study are available on request from the corresponding author.

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
