# Peer review of "Control of *Aculops lycopersici* with the Predatory Mite *Transeius montdorensis"

_insects, 2022, doi:10.3390/insects13121116_

Round 1

Reviewer 1 Report

Data seems to be of interest both to scientists and technicians/growers. However, a deep reorganization of the manuscript is mandatory as some parts in the material and methods are unclear and the presentation of the results should be revised.

Introduction

Lines 65-66: It could better refering to the Type III, subtype IIIb predator. See McMurtry,  de Moraes, Sourassou 2013, Systematic and Applied Acarology, 18(4): 297-320 http://dx.doi.org/10.11158/saa.18.4.1

Line 73: Do you mean astigmatid mites? Please, report here the species

Material and methods

Line 81: on tomato plants

Line 92: What means section? Parts of leaves? How many sections/plant? Please, specify

Line 94: In the period of 3 months, plants formed only 6-8 leaves? Somewhat is wrong!

Line 96: why only three samplings if predators releases lasted 3 months?

Line 97: The plants were cut…. This part of the experiment is unclear. How many plants were cut per sampling?

Line 115: replicates

Line 115: … treatment. For what you wrote six replicates per treatment correspond to 60 plants per treatment. Is it so? Please specify.

Line 117-118: (in similar number to ….). This means about 170 astigmatid prey/plant or better predator/prey ratio=1:1. Why on potted plants you used 140,000/170 predators that is 1:824?

Line 124: How many samplings you carried out during the observation period? Please, report this datum

Line 125: Three plants out of 60, represent 5% of plants. It is ok in field samplings but here it seems to be a small sample.

Line 128: What do you mean with upper and lower leaves? The upper and lower surfaces of leaves or leaves from the upper and lower part of the plant? In the second case, please give information on the ordinal number of sampled leaves.

Lines 128-129: Please, explain better this sampling technique. Do you adopted sticky tape imprints to entrap mites that were present in the instant of sampling on the petiole junctions or you left the sticky tape around of the petiole for a period of time? In any case how many sticky tape sections per sampling?

Line 132: The top of the plant is a vague reference. Please, specify height and also the ordinal number of leaves

Lines 136-137: This part is unclear. How you defined the damage limit? Please report the method you adopted.

Line 143: Autumn starts at 21 of September, so you should redefine the phrase or keep only the period: from July to December. Similarly for Spring (March 21, June 21)

Line 168: I’m really curious to know why the authors adopted this extraordinary mite-day. Which is the practical use of this measure? Please, furnish a scientific justification for this method.

Line 171: The normal distribution of data are needed prior to Student’s t-test. Are you assessed it? Have you transformed raw data before statistical analyses? If the normal distribution don’t met, you can use a very useful non parametric method (NPC) to compare data collected in successive time intervals (see Higgins 2004 and Pesarin 2001). The software is free: http://static.gest.unipd.it/~salmaso/NPC_TEST.htm

Line 195: .. very few TRM… Very few? You reported no one!

Line 199: Figure 2.   Are there significant differences between control and T. montdorensis treatments for these parameters? Please, report significance for each sampling date.

Line 206: … were found in the control than… For each date or for the whole period?

Line 217: ET… What means?

Line 220: Figure 3. There is some wrong in this figure. On the left you reported both A. lycopersici and T. montdorensis with a same scale of mite-days. Basing on this figure, in the last sampling for 800 specimens of A. lycopersici should be about 70 specimens of T. montdorensis in the sticky tape sampling, and for about 1,6 millions of A. lycopersici about 150,000 specimens of T. montdorensis. These numbers are absent in the text and seem factitious.

Moreover, what mean A and B in the first figure should be mentioned in the caption.

Finally, in materials and methods you reported 5 destructive samplings, but in the figure only 4 are reported. Why?

Line 229: Figure 4. Please indicate what means the asterisks

Lines 235-243: This part should be moved to Materials and methods chapter

Line 268-269: If it is not statistically different has not a similar trend!

Line 274: Please, check the value of P

Discussion

No one word in this chapter about the prey for the phytoseiid mite, Suidasia medanensis. This acarid has been reported as a very allergenic mite with increasing incidence in high relative humidity environments. No word deserves the health of agricultural workers? Could this method be considered of low impact? In my opinion, authors should discuss about this topic. Not only synthetic chemicals are dangerous for the human health.

Lines 305-307: The predator was able ….Figure 4). What about the results obtained in the previous test? Are not contradictory? How you explain it?

Line 322: Please, move the highlighted part to the Results

Discussion needs to be improved

See some remarks in the revised text

Author Response

REVIEWER 1.

- Lines 65-66.- It could better refering to the Type III, subtype IIIb predator. See McMurtry,  de Moraes, Sourassou 2013, Systematic and Applied Acarology, 18(4): 297-320 http://dx.doi.org/10.11158/saa.18.4.1. We don’t see the point of going into such a detail of the type of predator used. This is not a study on laboratory predator behavior but about its use as a biocontrol agent.

.- Line 73.-Do you mean astigmatid mites? Please, specify the species. The prey species used is not as important as for being cited in the definition of the hypothesis. Nevertheless, the astigmatid species was mentioned in line 83 of the M&M, when we specify the details of the study.

-Line 81.-“On” tomato plants. Done.

-Line 92.- What means section? Parts of leaves? It is already mentioned sections of “stems”, no leaves involved. 1-3cm added How many sections per plant? Please, specify Added “per plant” line 93.

- Line 94.-In the period of 3 months, plants formed only 6-8 leaves? Somewhat is wrong! Nothing is wrong. Plants kept in a climatic chamber at 25ºC and with the artificial light intensity of the chamber do not grow as they would do in a greenhouse.

- Line 96.-why only three if predators releases lasted 3 months? With 8 plants per treatment, we could not destroy them every week and have a significant number of replicates in each sampling date!. Therefore, we did a first wash of just one plant on the 4th of June to see if we were able to detect the predator and the TRM at low level. Then we did two more washings of 3 plants each on the 19th June and 3rd July, when populations of TRM and the predator were high and when they were almost gone.

- Line 97.-This part of the experiment is unclear. How many plants were cut per sampling? As mentioned in the previous response one, three and three. We had added this info in lines 98-99 as also requested by reviewer 2.

- Line 115.-“replicates”. Done. -That means 60 plants/treatment? Please, specify. For what you wrote six replicates per treatment correspond to 60 plants per treatment. Is it so? Please specify. Since we used a randomized block design to account for the variability of the greenhouse sections, the replicates were the cages not the plants. This is the appropriate experimental design commonly used in cage experiments that simulate the crop. The mean of three plants was used as the measure of each cage for comparisons.

- Line 118.-This means about 170 prey/plant or better ratio predator/prey 1:1? Why on potted plants you used 140,000/170 predators? Why on potted plants you used 140,000/170 predators that is 1:824?. As mentioned to reviewer 2, we have added the prey quantity in lines 119 and 124, since this was not well explained in the manuscript. In the potted plants we tested a 1:824 pred: prey ratio and in the cages experiment we tested a 1:1200 ratio, which was higher. Nevertheless, we were not interested in testing any specific pred:prey ratio, we wanted to test if the strategy could work for the control of TRM or not.

-Line 124: How many samplings did you carry out during the observation period? Please, report this data. Done in line 127.

-Line 125: Three plants out of 60, represent 5% of plants. It is ok in field samplings but here it seems to be a small sample. We have sampled three plants out of ten in each cage, as mentioned before, and this is 30 % of the plants sampled.

- Line 128.-What do you mean? The upper and lower surfaces of leaves or leaves from the upper and lower part of the plant? In the second case, please give information on the ordinal number of sampled leaves. We refer to the leaf above and the leaf below the damage limit of collapsed trichomes, that is two leaves per plant. We inspected the whole area (upper and lower surface) of these two leaves: “top and bottom sides inspected” was added in Line 130.

-Lines 128-129: Please, explain better this sampling technique. Do you adopted sticky tape imprints to entrap mites that were present in the instant of sampling on the petiole junctions or you left the sticky tape around of the petiole for a period of time? In any case how many sticky tape sections per sampling? We stick tape sections and removed right away, trapping mites present in that moment. “and removing” was added. Two “tape sections per plant” also added. Line 131.

-Line 132: The top of the plant is a vague reference. Please, specify height and also the ordinal number of leaves. Damage was evaluated at the limit where collapsed trichomes arrived, which was normally at the top of the plant. As plants grew up the number of leaves changed, therefore, to determine this limit by specifying the leaf number is not suitable. We now specify this “where the damage limit was located” in line 136.

-Lines 136-137: This part is unclear. How you defined the damage limit? Please report the method you adopted. It is already defined in line 103: “the maximum height at which collapsed trichomes were found”. As also explained it was observed visually or by using a magnifier lens if needed. We have added the expression “damage limit”, line 105.

-Line 143: Autumn starts at 21 of September, so you should redefine the phrase or keep only the period: from July to December. Similarly for Spring (March 21, June 21). Added “cycle” for better precision. Lines 147-148.

-Line 168: I’m really curious to know why the authors adopted this extraordinary mite-day. Which is the practical use of this measure? Please, furnish a scientific justification for this method. This is an old and common way for accounting the total effect of mites (or insects) between two sampling dates. Consists in multiply the mean of scores of two consecutive sampling dates by the number of days between these samplings. It better describes the effect of arthropod pest on plants than just the number of individuals of an specific sampling date. See for example [5] Royalty, R.N.; Perring, T.M. Reduction in photosynthesis of tomato leaflets caused by tomato russet mite (Acari: Eriophy-idae). Environ. Entomol. 1989, 18(2), 256–260. DOI 10.1093/ee/18.2.256.

-Line 171: The normal distribution of data are needed prior to Student’s t-test. Are you assessed it? Have you transformed raw data before statistical analyses? If the normal distribution don’t met, you can use a very useful non parametric method (NPC) to compare data collected in successive time intervals (see Higgins 2004 and Pesarin 2001). The software is free: http://static.gest.unipd.it/~salmaso/NPC_TEST.htm. Thank you for the recommendation. Of course, normal distribution was assessed in all tests and proceed accordingly transforming data, if needed. When data did not meet normality and homocedasticity of variances we used the non-parametric Student-t test which reduces the number of degrees of freedom. We think that is not necessary to specify all these basic procedures in the manuscript.

-Line 195:  very few TRM… Very few? You reported no one! corrected. Line 201.

-Line 199: Figure 2.   Are there significant differences between control and T. montdorensis treatments for these parameters? Please, report significance for each sampling date. We did not made any statistical comparisons of these parameters in the potted plants experiment since they were just indicatives of the spread of the TRM population; the statistics were performed with the data of the sticky tapes shown in figure 3.

-Line 206: … were found in the control than… For each date or for the whole period? In the last samplings, added lines 211.

Line 217: ET… What means? Changed for SE.

-Line 220: Figure 3. There is some wrong in this figure. On the left you reported both A. lycopersici and T. montdorensis with a same scale of mite-days. Basing on this figure, in the last sampling for 800 specimens of A. lycopersici should be about 70 specimens of T. montdorensis in the sticky tape sampling, and for about 1,6 millions of A. lycopersici about 150,000 specimens of T. montdorensis. These numbers are absent in the text and seem factitious. The figure has been corrected as suggested also by reviewer 2, and now there is no confusion between treatments and T. montdorensis population. Moreover, what mean A and B in the first figure should be mentioned in the caption. Done.

-Finally, in materials and methods you reported 5 destructive samplings, but in the figure only 4 are reported. Why? Because here we are reporting mites per day and, as explained before, for 5 samplings you get 4 periods between samplings.

-Line 229: Figure 4. Please indicate what means the asterisks. Done.

-Lines 235-243: This part should be moved to Materials and methods chapter. These treatments were done for controlling different pests and diseases that attacked each crop and were not planned. Therefore, we believe are better suited in the Results section.

-Line 268-269: If it is not statistically different has not a similar trend! Correct, we have changed this expression, lines 276-277.

-Line 274: Please, check the value of P.  Yes, corrected.

-Discussion

No one word in this chapter about the prey for the phytoseiid mite, Suidasia medanensis. This acarid has been reported as a very allergenic mite with increasing incidence in high relative humidity environments. No word deserves the health of agricultural workers? Could this method be considered of low impact? In my opinion, authors should discuss about this topic. Not only synthetic chemicals are dangerous for the human health. Yes, we have included a last sentence in the Conclusions indicating the possible problem that this species can cause.

-Lines 305-307: The predator was able ….Figure 4). What about the results obtained in the previous test? Are not contradictory? How you explain it?In the previous test, the one on potted plants, plants were small and once the TRM arrived to the top the predator could not reverse this fact. But, in the cages experiment we had full grown plants in a greenhouse and the spread of TRM had a long way to go from the bottom to reach the top of the plant, and then the effect of the predator was more evident.

-Line 322: Please, move the highlighted part to the Results. The way we express this paragraph was confusing, as also mentioned by reviewer 2, and we have rephrased it for better clarity.

- Figure 5.-No mention on phytosanitary treatments against A. lycopersici are reported in the text. In lines 235-243 those of the autumn cycle and in lines 244-252 those of the spring cycle.

-Discussion needs to be improved. Done.

Reviewer 2 Report

The paper relates three series of experiments done in pots, cages and greenhouse for testing the release of Transeius montdorensis to control Aculops lycopersici on tomato crops. The studies are well described, the results interesting and the work done of a good quality (and high quantity of work for mite countings). I thus recommend to publish this paper, after minor revisions.

All my suggestions are included directly in the manuscript. They mainly concern

- some english improvement

- graphs should be of better quality and the titles of the modalities renamed for avoiding confusion

- some information are lacking and should be added, and/or I did well understood some elements

- the discussion can be improved concerning the tomato variety, the repeated releases of predators and their alternative preys for instance

Author Response

REVIEWER 2.

All the English improvements made by the reviewer were accepted in the new version of the manuscript.

-line 67.- in all Europe?. Yes, T. montdorensis is commercialized in almost all of Europe. Agrobío is selling them in Greece, Rumania, Bulgaria, Finland, Sweden, Germany, Poland, Russia, UK (greenhouses only), France, Belgium, Holland, Spain, Portugal, Italy. In Switzerland and Croatia, they are not yet selling them yet, but they are preparing the documentation. It is sold by the companies Koppert, Biobest y Bioline Agrosciences. It is listed in the EPPO, and it was accepted because it is sold in at least 5 countries for more than 5 years.

-line 81.- what does it mean? Neighbor of what? Where from Spain? which region and locality? Done, we have specified the locality where the initial colonies were taken. line 81.

-line 97.- Two plants destructed at each date? Done, we have mentioned the number of plants examined in each destructive sampling. Lines 97-98.

-line 103.- predator's number counted also? No, predators were not counted since they moved too fast, and the visual counting was unreliable for estimating their density. We only annotated the top height on the plant at which we could see them moving. We have simplified the sentence for clarification.

-line 106.-??  experiments in cages?? better title? Done and changed for the titles of the other experiments, lines 86, 107, 145.

-line 116.-?? plantation?? age of the tomato plants? Done, 4 weeks old plants, line 111.

-Line 119. After reviewing the data, we realized that the real number of predators released was 350 instead of 175. This was like this because at that time Agrobio had improved their rearing and the density of predators was doubled.

-Line 123. The density of TRM introduced was also lower, 100 individuals aprox. per plant. It is corrected in the manuscript.

-lines 118 and 123.- it means 175 specimens which quantity of preys?? No, the amount of prey was much higher than the number of predators. The producer indicates that the density of prey was of 10000/ml with all developmental stages mixed. This was now included in the text.

-line 130.- how many tapes for each plant and each height? Two tapes per plant from the leaves above and below the height of TRM damage. There was only one height that was scaling up the plant as the TRM infestation progressed. Modified in line 131 adding “these” instead of “the” leaves. “In total, two tapes per plant were collected at each sampling date” was added in line 132.  

-line 132.- This methodology seems OK for A. lycopersici but for T. montdorensis such sampling is sufficient to assess the density of this predator? Yes, we agree with the reviewer that this method is not very precise for assessing the density of the predator. But this will be the type of sampling that a technician could do when releasing the predator in a commercial greenhouse. The rationale behind using this sampling methodology in our experiment was to assess the presence of the predator more than its density. The efficacy of the predator releases would be evaluated by the reduction of TRM density rather than by the abundance of T. montdorensis.

-line 140.- This technique is OK for A. lycopersici because of the difficulty to count such tiny mites, but for Phytoseiid I wonder about the representativity of the countings. We understand the concern of the reviewer. But, revising the manuscript we realized that there was a mistake in the Y axis titles of the second graph of Figure 1, meaning that we detected population levels of T. montdorensis of 70 to 370 while TRM were just 0 to 8 individuals. With the same technique, in Figure 3 we were able to detect up to 3000 individuals of T. montdorensis. Therefore, it gave us sufficient data to assess the presence and abundance of both mites in the system.

-line 146.- not the same variety as in other experiments. At least, this should be discussed as we know the impact of variety on predator dispersal. We agree, we have included it in the discussion lines 313, 354-356.

-line 151.- which region? Almeria, Spain? No, we refer to the area around Barcelona, where the trials were performed. Included in Line 154.

-line 156.- quantity known? The quantity of prey added (112500 and 210500) was incorporated into lines 159 and 161.

-line 157.- why a difference in the number of releases between spring and autumn? The autumn cycle was shorter than the spring one (Figure 5) due to weather conditions since these were not heated greenhouses. As the season progressed, temperatures and photoperiod in the autumn cycle were not favoring the TRM development while it was the opposite in the spring cycle. For this reason, the number of predator releases were higher in spring than in autumn. We tested T. montdorensis in these two crop cycles to see the robustness of the strategy, since greenhouse tomato is produced in our area (around Barcelona) in these two seasons.

-line 174.- verbs is lacking in the sentence. Done, sentence changed by: “data from the sticky tape samplings in the experimental greenhouse experiment was Also compared by this test”. Lines 177-178.

-line 182.- / plant / ml? You observed aliquots and then you estimated the quantity by plant? Yes, the counting of the 10 aliquots (1ml each) were averaged and multiplied by 60ml, the amount of water+ wetting agent used. We have added “estimated“ in the sentence. Line 186.

-Figure 1.- If I well undestood, for each date you have two plants, so I think that standard error are not appropriate, I suggest the report the two values per date on the graph. In fact, as specified now in the text, we had one plant on the first sampling date and three in the second and third sampling dates. Therefore, standard error is correct for the two last samplings but not for the first one, which we had deleted it from the graph.

-line 196.- why? they would stay in the bottom on the plants if there is also no prey there also? We have deleted the last part of the sentence where it is said that predators were found on the bottom of the plant. We don’t know why they were there since prey was lacking on all the plant.

-Figure 2.- same remark as above concerning standard errors. Since we did the measurements on 8 plants per treatment, this allows us to properly calculate the standard errors. We have specified the number of plants measured in M&M, line 103.

-line 207.- I do not understand this sentence. Modified for clarification. Also, Figure 3 has been modified according to reviewer’s suggestion, with a different name for treatments together with a different line design.

-line 214.- than what? Sentence corrected, line 218.

-line 215.- what about statistical differences in predator densities? No statistical differences were observed in the washed plants because the variability was very high. Probably because the predator densities were low at the last sampling date (29 of January, Figure 3) and in one of the plants of the predator treatment the TRM were increasing again.

-line 217.- But your destructive sampling in november was done on two or four leaves not on all the plant??? Yes, correct the sampling was done in four leaves of the plant. I do not understand here the relation between the number released / plant and the number sampled / leaf ??? We have added (line 220) that the number of mites-days was estimated per four leaves, for clarification.

-Figure 3.- The legend is not clear on the left graph, please replace T. montdorensis by "modality with introduction of the predator" Done. The reviewer is right, there is confusion between the treatment names and the predator’s name. We have named treatments as “Control treatment” or “Predator treatment” and changed it accordingly in all figures. “T. montdorensis” was only used in legends for indicating the density of predators.

-line 259.- better explain how this conforms the previous results? We have rephrased the paragraph and now is better explained. Lines 262-265.

-line 269.- Similar? Yes, done. Line 274.

-line 280.- These differences are only due to the control of Aculops lycopersici or other pests / deseases as the two modalities did not receive exactly the same treatments ??? Pests were similar in both treatments in the autumn cycle, since only one treatment was done at the beginning of the cycle in both plots for Helicoverpa armigera and another one for Liriomyza spp. (one treatment in the control plot and a release of Diglyphus in the predator plot). In the spring cycle no other pests than Aculops needed any treatment, only powdery mildew needed to be treated with sulphur. So, we think that the main yield differences were not due to the incidence of other pests /diseases.

-line 291.- Transeius montdorensis was descrbed by Schicha on Datura sp., and some specimens were also found on tomato feeding on Eriophyd species see: http://www.lea.esalq.usp.br/newphytoseiidae/web/busca/especie/detalhe/transeius-montdorensis-schicha-1979. In Steiner et al. 2003. they described: “ The holotype was from Datura leaves at Mount Dore. Further collections were made at the time from tomato (feeding on eriophyid mites) and from Mucuna sp. Schicha (1979) lists its distribution as Queensland, Fiji, New Caledonia, New Hebrides (Vanuatu) and Tahiti. Host plants are listed as Ageratum sp., Cucumis sativa (cucumber), Datura sp., Eucalyptus sp., Fragaria annanassa (strawberry), Lycopersicon esculentum (tomato), Mucuna sp., Oxalis sp., Phaseolus atropurpureus (purple bean), Phaseolus vulgaris (green bean), Sechium edule (Choko) and Sida acuta.” We have added “and other plants” line 296.

-line 294.- be careful, as this depends also on the variety (dispersal on tomatoes). May be Schicha found T. montdorensis on a not pilose variety (with gandular IV trichomes)? It is not because it is reported on tomato that it is adapted to. Many factors can affect the occurrence. We agree to soften this sentence since different tomato varieties could have different density of trichomes. Line 296.

-line 301.- comment on the tomato variety is expected as not the same variety was used in the different experiments.  Comment added. Line 307.

-line 310.- do you think that astigmata were still present? Yes, the rearing prey was present since we release it fortnightly until the 20th of December. How the alternative prey presence might also impact the predator dynamics? The alternative prey is what allows to maintain and grow the predator population. and preference for the other preys? There were no other prey for the predator than TRM and Astigmata. This should be discussed as the originality of this study is (i) repeated release of T. montdorensis and (ii) release of T. montdorensis with huge quantity of alternative prey. Included a new sentence in Lines 310-312.

line 320.- be careful as yield can also be affected by other pests, the management of the two modalities being different?. Included a new sentence in Lines 322-323.

- line 330.- which ones? TRM?. be careful, this not really a discussion as results are still presented here. Maybe this was not clearly expressed since we were discussing the results of other authors. We have rephrased these sentences for clarification, Lines 327-331.

- line 340.- I think that this part can be reduced as not directly associated to the results obtained. Yes, we have suppressed this sentence since it was not directly observed by us.

- line 343.- in case of an attack of T. urticae?? We have also suppressed this sentence and the citations.

- line 348.- I think that more perspectives should be added, especially concerning the survival of T. montdorensis when alternative preys disappear, concerning the amount of releases (and costs) and the competition between alternative preys and TRM. We have included a last sentence addressing these suggestions.

Round 2

Reviewer 1 Report

Dear authors,

thank you for your answer to my questions. However, I noticed a certain haste in making the proposed changes, that seems far from the spirit of collaboration that guided my remarks.

I think that your data are robust and useful for readers but I think also that in more parts this manuscript is still confused and this certainly reduces the impact this work can have in the scientific field.

My best regards

Author Response

Thank you for all the suggestions made for improving this manuscript, we really appreciate them since they helped to clarify unclear points. This has been a laborious manuscript with many details to be improved and the different points of view provided by reviewers were very important for detecting inconsistencies that we, the authors, were not able to see. We are sorry if we were too direct on some points, and we appreciate that your final opinion was positive.

With our best regards.

Cristina Castañé, the corresdponding author.